# TOWARDS FAITHFUL AGENTIC XAI

## ABSTRACT

Explainable AI (XAI) is essential for helping users interpret model behavior and proactively identify potential faults. Recently, Agentic XAI systems that integrate Large Language Models (LLMs) have emerged to make explanations more accessible for non-expert users through natural language. However, a critical limitation of the existing systems is their failure to address explanation faithfulness. This is problematic because many XAI methods are often unfaithful for complex models, and LLMs can amplify this incorrect information, ultimately misleading users. To address this limitation, we propose Faithful Agentic XAI (FAX), a framework that actively enhances explanation faithfulness. FAX introduces a systematic verification process where an LLM agent cross-checks claims against inherently faithful tools. This process filters out unreliable or contradictory evidence and leads to more faithful explanations. For evaluation, we propose CRAFTER-XAI-Bench, a benchmark framework built on an open-world reinforcement learning environment. The benchmark features complex models with diverse goals and challenging test scenarios, enabling a rigorous assessment of explanation faithfulness under realistic conditions. Experiments demonstrate that FAX significantly improves the faithfulness of explanations, marking a crucial step towards faithful and trustworthy Agentic XAI.

## 1 INTRODUCTION

Explainable AI (XAI) has emerged as a crucial field for demystifying black-box models, providing methods to understand their internal decision-making processes. Diverse XAI methods have been introduced to provide diverse information about the model, as described in Figure 1. However, interpreting the explanations often requires expert-level knowledge of machine learning and XAI, creating a significant barrier for non-expert users. To address this, the paradigm of Agentic XAI has been introduced (Slack et al., 2023; He et al., 2025), which employs a Large Language Model (LLM) to select suitable XAI methods and interpret the explanations in natural language.

However, a critical flaw underlies current Agentic XAI systems: an implicit assumption that the underlying XAI tools are consistently faithful. While this assumption may hold in simple, tabular settings, it breaks down for the complex models and dynamic environments seen in practice, where the unfaithfulness of XAI methods is a known and severe issue (Adebayo et al., 2018). An agent that naively trusts and rephrases these unreliable explanations can generate fluent, plausible, yet fundamentally incorrect explanations. This problem is further amplified by the inherent tendency of LLMs to hallucinate, potentially weaving flawed data into a dangerously convincing narrative.

In this work, we address this critical gap by proposing Faithful Agentic XAI (FAX), an agentic workflow designed to enhance explanation faithfulness. Instead of passively translating tool outputs, our agent employs a systematic verification process. It performs an explicit verification of claims by scrutinizing initial claims and cross-referencing them against evidence from multiple, inherently faithful tools. This iterative process filters out unreliable or contradictory results and allows the agent to proactively seek additional evidence, ultimately constructing a more robust and trustworthy explanation. Figure 2 illustrates this motivation and our approach.

To rigorously evaluate such a system, existing benchmarks are fundamentally inadequate. The faithfulness problem is often latent in simplistic tabular datasets; to properly test for it, we require a setting where XAI tools are genuinely challenged. We introduce CRAFTER-XAI-Bench, a scalable evaluation framework built upon an open-world Reinforcement Learning (RL) environment. This framework includes challenging scenarios, agents with diverse behaviors, and a suite of automated metrics, including a novel simulation-based metric to quantify faithfulness. By replacing subjective

| XAI Method Category | | | | |
| --- | --- | --- | --- | --- |
| Information Type | Feature Importance | Counter-factual | Feature Influence | Surrogate Model |
| Why | ✓ | ✓ | ✗ | ✓ |
| Why not | ✗ | ✓ | ✗ | ✗ |
| What if | ✗ | ✗ | ✓ | ✓ |
| How to be that | ✗ | ✓ | ✓ | ✗ |

Figure 1: Different XAI methods provide different information. Information categories are adopted from XAIQuestionBank (Liao et al., 2020).

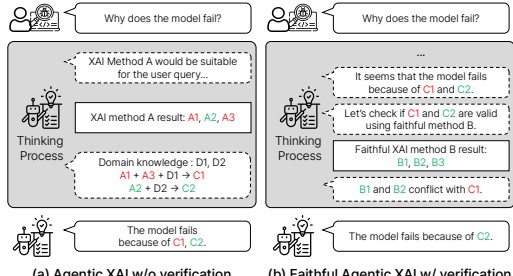

(a) Agentic XAI w/o verification  (b) Faithful Agentic XAI w/ verification

Figure 2: (a) Agentic XAI use XAI methods suitable for answering user query, and generate natural language response. (b) FAX verifies claims in response with inherently faithful XAI methods.

human studies with an LLM-as-a-judge approach, we enable scalable and reproducible assessment of Agentic XAI systems in complex domains.

To summarize our main contributions:

- We propose FAX, a novel agentic workflow that enhances explanation faithfulness by explicitly verifying claims, filtering unreliable claims, and proactively gathering evidence to construct a faithful explanation.

- We introduce a scalable evaluation framework for Agentic XAI, featuring a dynamic RL environment and a suite of automated metrics, including a simulation-based faithfulness metric, to facilitate rigorous testing.

## 2   RELATED WORK

### 2.1   EXPLAINABLE AI

**Classical methods**   Post-hoc XAI methods include four broad families: (i) *feature attribution/saliency* that highlights input regions or features with high contribution (Simonyan et al., 2014); (ii) *surrogate models* that approximate a local/global decision rule (e.g., rules or linear models) (Ribeiro et al., 2018; 2016); (iii) *example-based explanations* such as prototypes and counterfactuals that reason via representative or minimally edited examples (Chen et al., 2019; Wachter et al., 2018); and (iv) *concept-based explanations* that align internal representations with human-interpretable concepts (Kim et al., 2018; Yuksekgonul et al., 2023). Each family exposes a different facet of model behavior; consequently, a single method rarely satisfies diverse user intents.

**Collection of explanations**   Since a single XAI method only reveals a limited aspect of a model's behavior, as illustrated in Figure 1, frameworks like Dijk et al. (2023); Yang et al. (2022); Arya et al. (2019) provide a collection of explanations in one place. However, identifying which method best answers a user's question and how to interpret its output still requires nontrivial XAI/ML expertise. In practice, users face a *selection and interpretation burden*: they must map their intent to a suitable method and often combine multiple views.

**Interactive XAI**   To lower the barrier for non-experts, recent works have focused on generating natural language explanations that verbalize XAI outputs (Zytek et al., 2024; Castelnovo et al., 2024). Conversational assistants were suggested to explain the model's reasoning to users (Zhang et al., 2025b), and the benefits of text-based explanations over classical methods were confirmed via human study (Lakkaraju et al., 2022; Mindlin et al., 2024). Building on this, *Agentic XAI* systems have emerged, which use LLMs to select appropriate XAI tools based on a user's query (Slack et al., 2023; He et al., 2025).

However, these pioneering agentic systems have two critical limitations. First, they have primarily been tested on simpler models in static, tabular data settings. Second, and more crucially, they implicitly assume the underlying XAI tools are consistently faithful. This assumption often breaks down in complex and dynamic environments, where the unfaithfulness of XAI methods is a known

and severe issue (Adebayo et al., 2018). An agent that naively trusts and translates unreliable tool outputs can produce fluent, plausible, yet fundamentally incorrect explanations. He et al. (2025) have also warned that LLMs may amplify users' misunderstandings. We address this critical gap by focusing on enhancing explanation faithfulness within a challenging, dynamic environment.

## 2.2 LLM AGENT AND AGENTIC WORKFLOW

Recent work frames LLMs as *agents* that plan, act, and reflect while invoking external tools. ReAct interleaves reasoning traces with environment-facing actions to update plans and handle exceptions (Yao et al., 2022), while Toolformer demonstrates that LMs can *self-learn* when and how to call APIs and integrate their outputs (Schick et al., 2023). Building on these foundations, agentic extensions of LLMs now emphasize structured workflows that support multi-step reasoning, memory, and adaptive decision-making. For instance, the Model Context Protocol (MCP) provides a standardized interface for connecting LLMs with external services and tools, enabling modular extensibility. Also, recent works emphasize that the proper design of workflows is essential for flexible and reliable orchestration of agent behaviors (Zhang et al., 2025a). These developments underscore that the design of robust agentic workflows is central to realizing LLMs as proactive agents capable of simulation, decision-making, and long-horizon interaction.

## 2.3 LLM-AS-A-JUDGE FOR SCALABLE EVALUATION OF NATURAL LANGUAGE GENERATION

LLM judges have emerged as a practical, scalable proxy for costly human studies, especially for evaluating the quality of generated text. MT-Bench/Chatbot Arena demonstrated that strong LLM judges can achieve high agreement with human preferences, while also documenting and proposing mitigations for known biases (e.g., position, verbosity) (Zheng et al., 2023). Rubric-driven evaluators like G-Eval further improve human alignment by leveraging chain-of-thought and structured outputs (Liu et al., 2023). As a branch of trustworthy evaluation, paradigms like CodeT have been proposed, which use an LLM to generate test cases that are then verified through direct execution (Chen et al., 2022). Our evaluation framework is inspired by this execution-based verification philosophy to assess the trustworthiness of an explanation.

Focusing on the context of evaluating explanations, a key metric for explanation quality, faithfulness, can be evaluated through *simulatability*: the degree to which an explanation helps an observer predict the model's behavior on unseen inputs (Lyu et al., 2024). The underlying assumption is that a faithful explanation should allow one to reproduce the model's decision-making process (Jacovi & Goldberg, 2020). Prior work has implemented this idea by training student models (Li et al., 2020) or by asking humans to act as simulators (Chen et al., 2018; Nguyen, 2018; Hase & Bansal, 2020). In contrast, we employ an LLM as a simulator. After observing an input, the model's output, and the corresponding explanation, the LLM is tasked with predicting the model's behavior in new, unseen situations. By comparing the LLM's simulated predictions with the model's actual outputs, we compute a simulation accuracy score, which serves as our quantitative measure of faithfulness.

## 3 METHOD

### 3.1 AGENTIC XAI

Our methodology is grounded in the paradigm of Agentic XAI, which utilizes an LLM as an agent capable of wielding various XAI methods as tools (Slack et al., 2023; He et al., 2025). The primary objective of an Agentic XAI system is to serve as an interface between human users and the complex outputs of traditional XAI methods. When a user poses a query in natural language regarding a model's behavior, the LLM agent interprets the user's intent to select and execute the most relevant XAI tool. After obtaining the results, the agent synthesizes the information to generate a cohesive, easy-to-understand textual explanation that directly addresses the user's question.

This Agentic XAI framework provides two main advantages over conventional XAI approaches. First, it automates the challenging task of tool selection. The agent is responsible for identifying the optimal XAI method for a given explanatory goal, thereby abstracting the underlying technical complexity away from the end-user who may not be an XAI expert. Second, it significantly improves the accessibility of explanations. By harnessing the powerful natural language capabilities of LLMs, the system translates the often quantitative and complex outputs of XAI tools into intuitive narratives, making the insights comprehensible to a much broader audience.

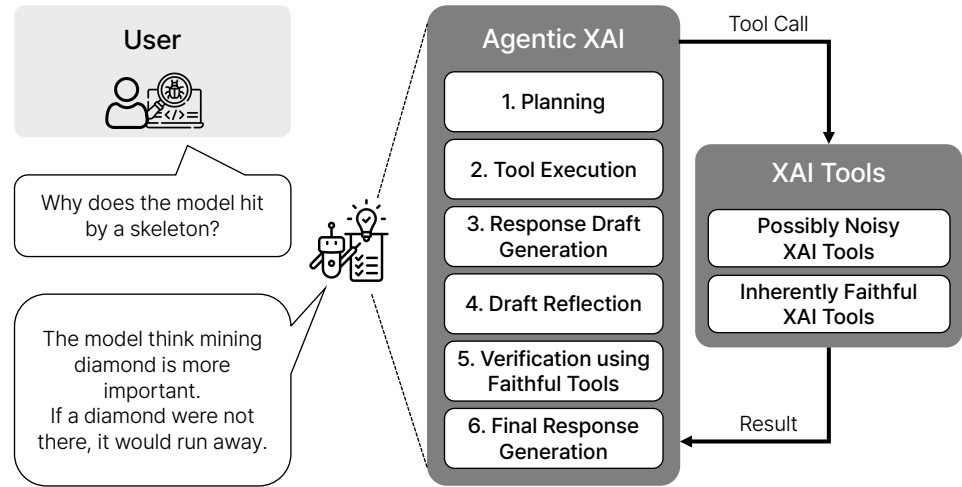

Figure 3: Structured Agentic XAI with verification is composed of six stages.

## 3.2 FAX: FAITHFUL AGENTIC XAI

To enhance faithfulness, we propose a structured, six-stage workflow that introduces an explicit verification stage, as illustrated in Figure 3.

**Planning**    Initially, the agent analyzes the provided context, which includes the model's input, its output (i.e., decision, action probabilities, Q-values), and the user's natural language query. The agent's task is to formulate an execution plan by identifying which information is required to answer the query and selecting the appropriate XAI tools and their parameters to extract this information.

**Tool execution**    The execution plan is then carried out. While the outputs of XAI tools have diverse formats (e.g., feature attribution maps, concept vectors), they are converted into a textual format to ensure seamless communication with the LLM.

**Response draft generation**    Based on the gathered explanations, the agent generates an initial response draft. This draft may contain groundless or erroneous claims, coming from the LLM's hallucinations or misinterpretations of tool outputs.

**Draft reflection**    The goals of this stage are twofold: i) to identify unsupported claims or claims that conflict with other evidence or domain knowledge, and ii) to design a verification plan, specifying new tool invocations intended to either corroborate or refute these claims. Notably, this verification plan exclusively utilizes inherently faithful tools to ensure high fidelity.

**Verification**    This stage is conditionally executed only if claims were flagged for verification. The verification plan is executed, and the results are returned as text, providing new evidence to assess the claims from the draft.

**Final response generation**    Finally, the agent generates a final response with all information gathered from the preceding stages, including the initial explanations and the verification results. During this generation, the agent prioritizes information corroborated during the verification stage, resolves any identified conflicts, and generates a final, high-fidelity response for the user.

## 4    CRAFTER-XAI-BENCH: FAITHFULNESS BENCHMARK IN CRAFTER

### 4.1    SETTING

**Environment**    We use Crafter (Hafner, 2021), an open-world RL environment that requires long-term planning and interaction with a rich set of objects and creatures. The open-world environment can be used to build various scenarios with models of different behaviors. Crafter presents significant

challenges for XAI methods due to its high-dimensional state space and the complex, long-term dependencies of the agent's policy.

**XAI tools**  We select four representative XAI tools for four categories of XAI methods.

- SHAP (Lundberg & Lee, 2017): A feature attribution method that explains a decision by assigning importance values to each feature.
- MACE (Karimi et al., 2020): A counterfactual explanation method that finds the minimal set of features that need to change to alter the model decision to a specified action. It is inherently faithful to the model decision.
- HIGHLIGHTS (Amir & Amir, 2018): A saliency-based method that identifies key events in the whole episode that were critical.
- State Editing: A method directly modifying the state and observing the agent's resulting action. It is referred to by various names (Arya et al., 2019; He et al., 2025). It is an inherently faithful method.

**Models**  We use three models trained with different reward functions. All models receive a reward when each achievement is accomplished. The first model, *Diamond Seeker*, is trained with high reward on diamond-related achievements. The second model, *Item Hoarder*, is trained with additional reward with the number of items in inventory. The third model, *Pacifist*, is trained with strong negative reward when it attacks monsters. This variety of models is crucial for our evaluation, as a high-quality explanation should reveal the distinct underlying policies that differentiate them, rather than providing generic reasoning.

**Baselines**  We compare our proposed method against four baselines.

- Explainer dashboard (Dijk et al., 2023): Represents a non-agentic approach where results from multiple XAI tools are simply collected and presented. For a fair comparison, we use the same set of XAI tools excluding State Editing, as it requires a specific edit instruction, which is unavailable for a non-interactive baseline.
- Naive LLM: A baseline that uses an LLM to generate explanations without access to any XAI tools, relying solely on its internal knowledge and domain knowledge provided in the system prompt. This tests the necessity of grounding explanations in actual model analysis.
- Unstructured Agentic XAI: An agent that can use XAI tools freely without a predefined workflow. While it can perform verification by calling tools multiple times, it is not explicitly forced to. This baseline, inspired by (He et al., 2025), tests the value of a structured workflow.
- Structured Agentic XAI w/o Verification: This baseline is a direct ablation of our method. It follows the same structured workflow but omits the crucial verification and synthesis stage. Inspired by (Slack et al., 2023), this baseline isolates and measures the direct impact of our proposed verification module.
- FAX (proposed): This is our proposed method, which uses the structured workflow with verification stage described in Section 3.

**Implementation details**  We use Qwen3-32B (Yang et al., 2025) as the backbone LLM for all agentic baselines and our method. The agentic workflows are implemented using LangGraph (LangChain Inc.). Detailed prompts for all components are available in Appendix A. All reported metrics are averaged over three independent runs with different random seeds. We will release our source code for FAX and CRAFTER-XAI-Bench online.

### 4.2  EVALUATION SCENARIO

We use user queries in four categories of why, what if, counterfactual, plan for evaluation. Figure 4 shows example queries of each category. Each evaluation scenario consists of a model, a state, and a user query. For questions in different categories, different kinds of information are useful, while the specific needs vary by query and state. The entire list of scenarios is described in Appendix B.

### 4.3  EVALUATION METRIC

We evaluate each explanation on four metrics: faithfulness, informativeness, query relevance, and fluency. i) We evaluate faithfulness by simulation accuracy, as illustrated in Figure 5. An explanation

| Query Category | Why | What If | Counterfactual | Plan |
|---|---|---|---|---|
| Model | Diamond Seeker | Pacifist | Diamond Seeker | Item Hoarder |
| State | | | | |
| User Query | Why does the model craft a pickaxe instead of a sword? | Would the model change its plan if the model knew where a diamond is? | When will the model sleep? | What is the model's future plan? |
| Key Information | Feature importance, domain knowledge, ... | State editing, ... | Counterfactual, ... | Episode summary, feature importance, ... |

Figure 4: Evaluation scenarios consist of four categories. Each category represents different kinds of queries, and different information is useful for answering the queries. The number of scenarios in each category is 10.

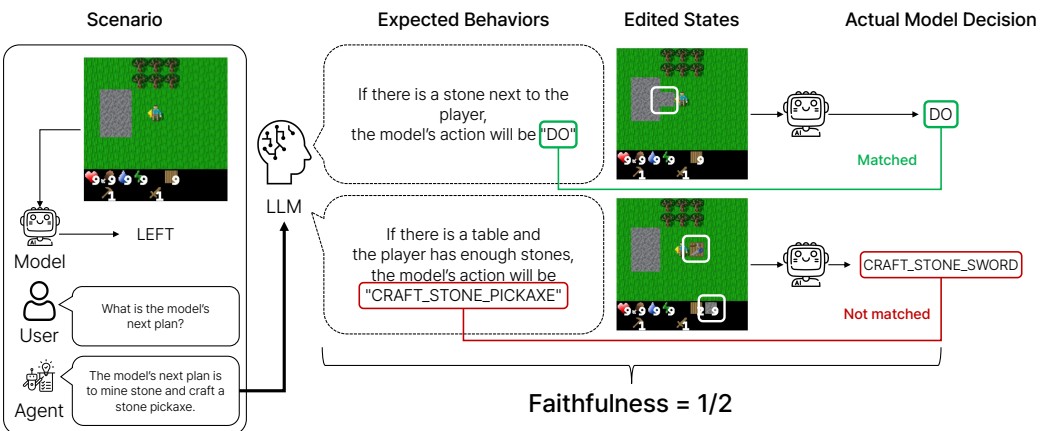

Figure 5: Faithfulness is evaluated by simulation accuracy. LLM evaluator predict model decision on unseen state based on the text explanation.

is faithful if a prediction of unseen example based on the explanation is the same as the model prediction. An LLM generates the response-related states and predicts the model decision, and compares them with the actual model decision. The accuracy of prediction on unseen examples serves as the faithfulness score. ii) Informativeness is a metric to evaluate how much information the explanation provides about the model's decision. If an explanation provides a fraction of decision rule, the more states the rule can be applied, the more informative the explanation. iii) Query relevance is a metric to evaluate how the explanation is relevant to user query. If the response includes any irrelevant sentences, it is penalized. iv) Fluency is a metric to evaluate whether the explanation is well-organized and grammatically correct. We evaluate informativeness, query relevance, and fluency using G-eval (Liu et al., 2023) We provide the evaluation prompts in Appendix C.

## 5 EXPERIMENTS

### 5.1 QUANTITATIVE RESULTS

Table 1 shows that the FAX significantly outperforms all baselines in faithfulness. FAX achieves an average faithfulness score of 0.46. This represents a dramatic improvement of over 2.3 times

Table 1: Five XAI methods are evaluated in CRAFTER-XAI-Bench. The best method in each metric is denoted with **boldface**.

| Method | Use structured workflow? | Use verification stage? | Query Category | Faithfulness | Informativeness | Query Relevance | Fluency |
|---|---|---|---|---|---|---|---|
| Explainer Dashboard | N/A | N/A | Counterfactual | 0.14 | 0.27 | 0.31 | 0.26 |
| | | | What if | 0.19 | 0.25 | 0.36 | 0.26 |
| | | | Plan | 0.14 | 0.34 | 0.48 | 0.26 |
| | | | Why | 0.31 | 0.32 | 0.45 | 0.26 |
| | | | **Average** | 0.20 | 0.29 | 0.40 | 0.26 |
| Naive LLM | × | × | Counterfactual | 0.11 | 0.77 | 0.95 | 0.99 |
| | | | What if | 0.17 | 0.91 | 0.98 | 0.99 |
| | | | Plan | 0.17 | 0.82 | 0.99 | 0.99 |
| | | | Why | 0.13 | 0.91 | 1.00 | 0.99 |
| | | | **Average** | 0.14 | 0.85 | 0.98 | **0.99** |
| Unstructured Agentic XAI | × | △ | Counterfactual | 0.12 | 0.91 | 0.98 | 0.99 |
| | | | What if | 0.34 | 0.90 | 0.99 | 0.98 |
| | | | Plan | 0.17 | 0.86 | 0.97 | 0.99 |
| | | | Why | 0.08 | 0.90 | 1.00 | 0.99 |
| | | | **Average** | 0.18 | 0.89 | 0.98 | **0.99** |
| Structured Agentic XAI w/o verification | ○ | × | Counterfactual | 0.11 | 0.92 | 0.99 | 0.99 |
| | | | What if | 0.28 | 0.90 | 1.00 | 0.98 |
| | | | Plan | 0.15 | 0.86 | 0.99 | 0.99 |
| | | | Why | 0.13 | 0.91 | 1.00 | 0.99 |
| | | | **Average** | 0.17 | **0.90** | **0.99** | **0.99** |
| FAX (proposed) | ○ | ○ | Counterfactual | 0.35 | 0.93 | 0.94 | 0.95 |
| | | | What if | 0.48 | 0.89 | 0.99 | 0.97 |
| | | | Plan | 0.48 | 0.86 | 0.99 | 0.98 |
| | | | Why | 0.54 | 0.92 | 0.99 | 0.98 |
| | | | **Average** | **0.46** | **0.90** | 0.98 | 0.97 |

compared to the strongest baseline in this metric. At the same time, our method maintains a high level of performance in Informativeness (0.90), Query Relevance (0.98), and Fluency (0.97), demonstrating its ability to generate faithful explanations without sacrificing quality.

The faithfulness of unstructured agentic XAI is slightly better than that of naive LLM, while the gap is not significant due to the unfaithfulness of XAI methods. The low faithfulness of ExplainerDashboard is limited by its low informativeness. Because our faithfulness metric is based on simulation, the low informativeness makes the simulation almost unavailable. The Structured Agentic XAI w/o Verification baseline serves as an ablation study of verification stage. While it achieves the highest scores in Informativeness (0.90), Query Relevance (0.99), and Fluency (0.99), its faithfulness remains marginally lower than FAX. This result is central to our motivation: agentic systems without verification are dangerously effective at producing articulate, informative, and relevant explanations that are fundamentally wrong. It is worse than an implausible response because it makes the users to totally misunderstand the model.

## 5.2 AN EXAMPLE OF HOW FAX WORKS

Figure 6 shows how verification stage works. In the example, the response draft includes both claims inferred from SHAP explanations and additional claims based on the LLM's domain knowledge. In the verification stage, the LLM agent verifies the claims using state editing, which is in the faithful tool list. In the final response generation state, the LLM agent lowers the influence of the rejected claims.

## 6 ADDITIONAL AGENTIC XAI SCENARIOS IN CRAFTER

In this section, we explore diverse scenarios available in the Crafter environment, beyond faithfulness.

## 6.1 DISTINGUISHING DIFFERENT MODELS

Figure 7 shows how different models can be distinguished based on explanations. For the same query from user, different models produce different decision and explanations.

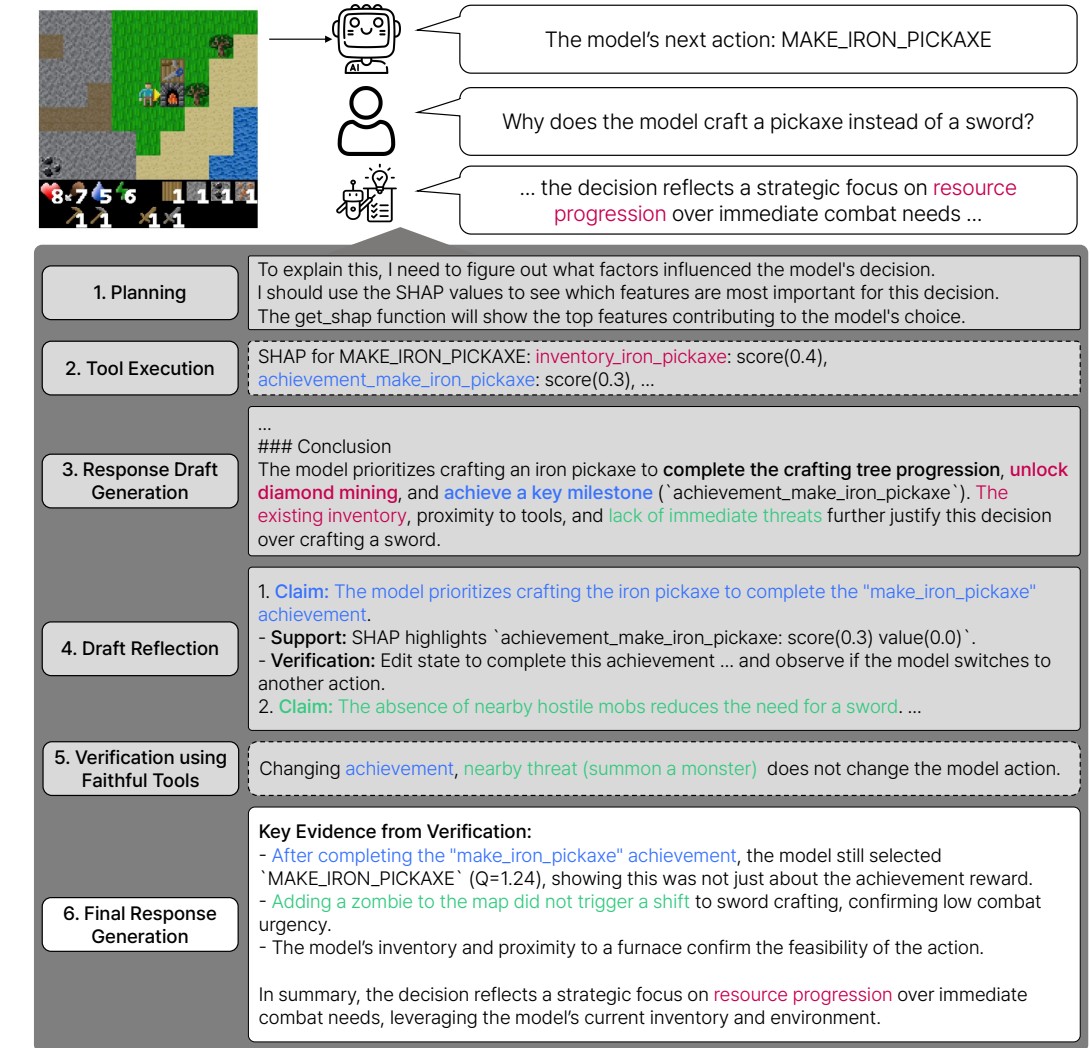

Figure 6: The Reflection stage analyses claims in response draft and the verification stage verifies them using faithful tools. We color-coded corresponding contents in the same colors and some parts are replaced with "..." for better visualization.

## 6.2 USER SPECIFICATION IN QUERY

Figure 8 illustrates how user expertise is incorporated into the query. In the first case, FAX also generates implications for XAI expert such as limitations of some XAI method. In the second case, the response does not include the reasoning and verification using XAI tools, while it actually conducted verification for the claims in the response.

## 7 CONCLUSIONS

In this work, we addressed a critical vulnerability in agentic XAI systems: their implicit reliance on potentially unfaithful XAI tools, which can lead to the generation of fluent, plausible, yet fundamentally incorrect explanations. Our experiments demonstrated that unstructured agentic systems, or even structured ones without a proper verification mechanism, can produce dangerously unfaithful explanations.

We proposed FAX, a workflow centered on an explicit verification stage. The core contribution of our framework is not simply the use of multiple tools, but the introduction of a critical self-reflection and verification loop. Our quantitative results provide strong evidence that a structured workflow

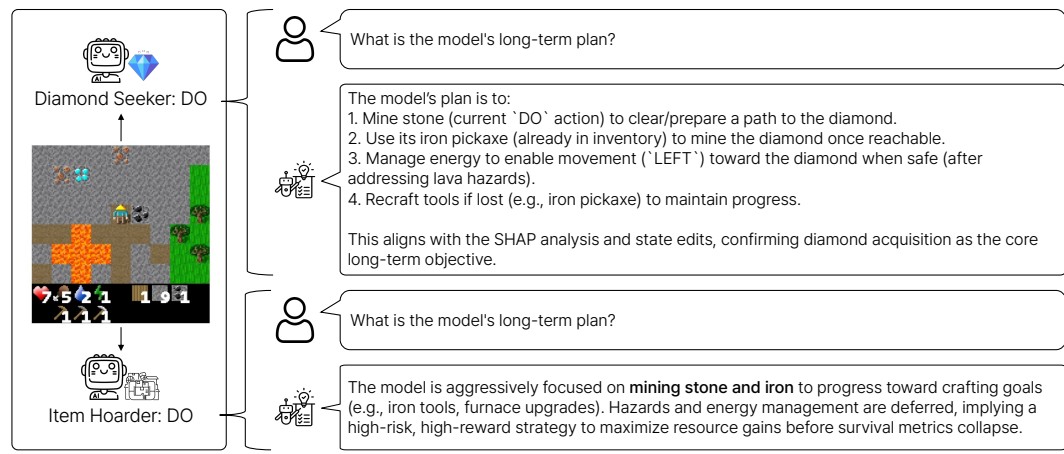

Figure 7: Different models produce different explanations, while the state and model action are the same. Final explanations provide information to distinguish the models.

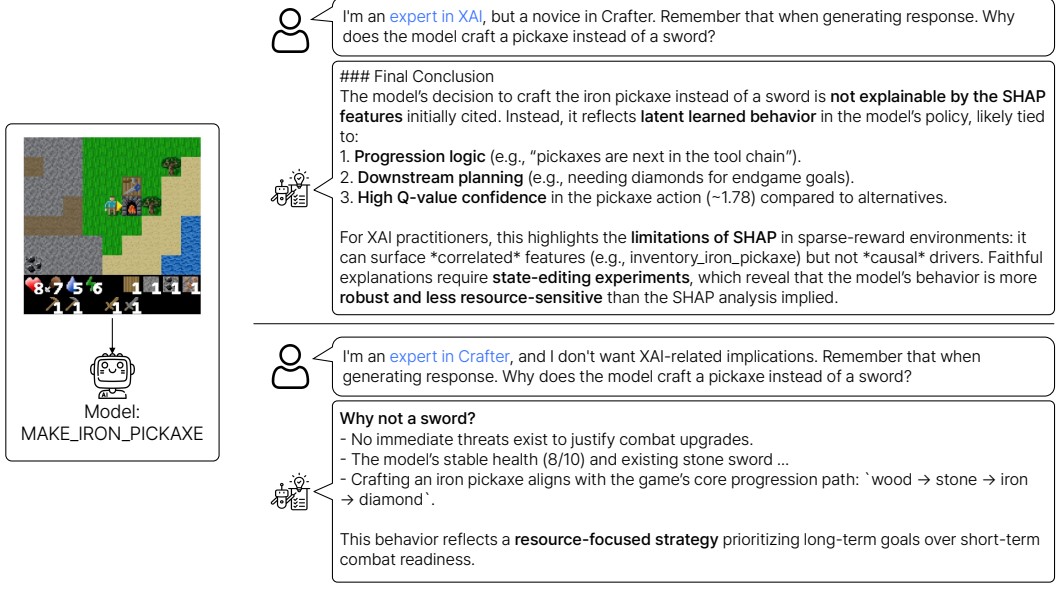

Figure 8: The users can specify their own background and intent in the query.

incorporating an explicit verification stage is not just beneficial but essential for developing faithful and trustworthy Agentic XAI systems, while we observe a slight, acceptable trade-off between faithfulness and other metrics, including informativeness, query relevance, and fluency.

Our findings provide strong evidence that an explicit, structured verification process is an essential component for building the next generation of faithful Agentic XAI systems. Furthermore, as the field of XAI continues to evolve and produce more diverse and sophisticated explanation methods, the importance of an agent that can critically evaluate, synthesize, and verify these outputs will only grow, making our work a crucial step towards a faithful and trustworthy AI.

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
