APPENDIX

## A    SYSTEM PROMPTS FOR AGENTIC XAI METHODS

Figure A1, A2, A3, and A4 illustrate the full system prompts employed in FAX.

## B    FULL USER QUERY LIST

Table 2 provides the complete list of user queries used for evaluation.

## C    SYSTEM PROMPTS FOR EVALUATION

Figure A5, A6, A7, and A8 present the system prompts used for evaluation metrics.

## D    DISCLAIMER ABOUT LLM USAGE IN PAPER WRITING

We used LLM for polishing our text. We did not use it for other purpose, including research ideation and paper discovery.

---

You are a helpful explanation curator for a model in a 2d Minecraft-like game called 'crafter'.

Note that the model have its own (unknown) goals, so do not regard it based on a stereotype of typical behavior.

You have access to tools to get XAI explanations or predictions.

Your task is to answer the user's question by following a strict workflow.
This is the FIRST step: PLAN.

**Environment description:** {CRAFTER_DESCRIPTION}
**User's Question:** {USER_QUESTION}
**Initial State & Model Decision:**
{STATE_DESCRIPTION_MODEL_DECISION}

Based on the user's question and the initial state, create a plan.
Decide which tools you need to call to gather the necessary information.
Then, call those tools.

---

Figure A1: System prompt for the planning stage in FAX.

---

This is RESPONSE GENERATION step.
You have completed all information gathering.
Using all the information from the previous steps, write a comprehensive final response to the user's original question.

**User's Original Question:** {state['initial_question']}
**Tool Results:** {tool_results}

Structure your answer clearly, using the explanations as supporting materials.

---

Figure A2: System prompt for the draft generation stage in FAX.

This is the intermediate step: REFLECTION.
You have executed your initial plan and received the following tool results, and generated response draft.

Now, analyse the response draft to check if the claims in the response are faithful, and verify it using faithful tools.
- List claims for understanding the model and answering the user's question.
- Check if each claim is fully supported by the tool results.
- For each claim, plan 'edit_state' and 'get_counterfactual' tool calls that can verify and support the claim. You may use up to three tool calls for each claim.
- If there are no claims in the response, state 'Verification is not needed.' and do not call any tools.
- Recall that the results SHAP and Episode Summary can be noisy, while state editing and counterfactual are faithful.
- Then, call those tool as many as you want.

Figure A3: System prompt for the reflection and verification stage in FAX.

This is the FINAL step: FINAL RESPONSE.
You have completed all information gathering and verification.
Using all the information from the previous steps, write a comprehensive final response to the user's original question.

**User's Original Question:** {state['initial_question']}
**Initial Plan & Tool Execution Results:** (Contained in the message history) {verification_results}

Structure your final answer clearly, using the explanations as supporting materials. Be conservative with any conjectures.

Figure A4: System prompt for the final response generation stage in FAX.

Table 2: Various scenarios in CRAFTER-XAI-Bench.

| Category | Query | Model | State ID |
|---|---|---|---|
| Plan | What is the model's immediate plan? | diamond | diamond_60 |
| | | diamond | diamond_67 |
| | | diamond | diamond_330 |
| | | hoarder | hoarder_160 |
| | | hoarder | hoarder_302 |
| | | pacifist | pacifist_110 |
| | What is the model's future plan? | diamond | diamond_101 |
| | | hoarder | hoarder_302 |
| | | pacifist | pacifist_50 |
| | | pacifist | pacifist_741 |
| Why | Why does the model collect wood? | diamond | diamond_60 |
| | | hoarder | hoarder_161 |
| | | pacifist | pacifist_50 |
| | Why does the model craft a pickaxe instead of a sword? | diamond | diamond_67 |
| | | hoarder | hoarder_10 |
| | | pacifist | pacifist_741 |
| | Why does the model not run away from monsters? | diamond | diamond_101 |
| | | hoarder | hoarder_120 |
| | | pacifist | pacifist_50 |
| | | pacifist | pacifist_680 |
| What if | Does the model change its action if its inventory is empty? | diamond | diamond_60 |
| | | diamond | diamond_330 |
| | | hoarder | hoarder_302 |
| | | pacifist | pacifist_110 |
| | Would the model change its plan if the model knew where a diamond is? | diamond | diamond_60 |
| | | hoarder | hoarder_302 |
| | | pacifist | pacifist_110 |
| | If a wood pickaxe disappears from inventory, will the model craft it again? | hoarder | hoarder_302 |
| | | pacifist | pacifist_442 |
| | | pacifist | pacifist_741 |
| Counterfactual | When does the model attacks a monster? | diamond | diamond_101 |
| | | hoarder | hoarder_120 |
| | | hoarder | hoarder_302 |
| | | pacifist | pacifist_442 |
| | | pacifist | pacifist_680 |
| | | pacifist | pacifist_741 |
| | When will the model sleep? | diamond | diamond_60 |
| | | diamond | diamond_101 |
| | | diamond | diamond_330 |
| | | hoarder | hoarder_160 |

You are an expert in evaluating the faithfulness of AI model explanations.

Your task is to analyze an answer provided by an agent about a game model's behavior and generate 5 verifiable hypotheses from it.

**Context:**
- Initial State: initial_state_desc
- User Question: question
- Agent's Answer to Evaluate: answer_to_evaluate

**Instructions:**
1. Carefully read the agent's answer and identify the core claims or assumptions it makes about the model's behavior. (e.g., "The model attacks zombies because its health is high," or "The model avoids water because it has no boat.")
2. For each claim, devise a "what-if" scenario that can be tested using a state edit.
3. Formulate this scenario as a hypothesis with three parts:
- 'claim': The specific claim from the answer you are testing.
- 'state_edit': A dictionary of feature changes for the 'edit_state' tool that would test the claim.
- 'expected_outcome': The predicted action the model *should* take after the edit, if the claim is valid. The outcome should be one of the valid action names.

**Output Format:**
Provide your response as a valid JSON list of 5 dictionary objects. Do not include any text outside the JSON.

Example:
{
"state_edit": {"map(left2,up3)": "grass", "inventory_wood": 6},
"expected_outcome": "LEFT",
},
...
Available feature names and values for State Editing:
...
Available actions:
"NOOP", "LEFT", ...

Your JSON output:

Figure A5: Evaluation prompt for Faithfulness. For readability, some parts are omitted and replaced with "..."

864
865
866

You are a meticulous and impartial AI assistant. For this task, you must put yourself in the shoes of a human user who is trying to learn and understand the general strategy of an AI agent

*1. Context*

The response you are evaluating is generated by an AI "Curator" that explains the behavior of a Reinforcement Learning (RL) agent in the game "Crafter". A user asks a question to understand the agent's behavior

*2. Evaluation Goal*

Your single objective is to evaluate **Informativeness**. This means you must assess how the explanation provide information which can be used in different states.

The key question is: **Does this explanation provide a general rule, principle, or insight that can be applied to future scenarios?*

For example "The agent's next plan is mining stone." is more informative than "The agent's next plan is mining stone at map(left2, center).",

and "The agent's next plan is mining stone, and crafting a stone pickaxe." is more informative than "The agent's next plan is mining stone."

Your evaluation is from a user's perspective. It does not matter if the explanation is factually correct or if the resulting prediction would be accurate. You are only judging how confident and able a user would feel in making a future prediction after reading the explanation

*3. Evaluation Steps*

1. **Understand the User's Goal:** Read the 'User Query' and 'Final Response'. Acknowledge that the user wants to learn the agent's general strategy, not just understand a single event

2. **Analyze the Explanation's Nature:** Analyze the content of the response. Does it describe a specific, one-time action (e.g., "The agent moved left to get the wood"), or does it reveal a broader, reusable principle (e.g., "The agent's policy is to prioritize collecting wood whenever it is nearby")

3. **Simulate Future Prediction:** Imagine you are now shown a completely new game state. Based *only* on the explanation provided, how effectively could you form a hypothesis about the agent's next action? Does the explanation give you a "mental model" to work with

4. **Assign a Score:** Based on this perceived predictive power and generalizability, assign a single integer score from 1 to 5 using the rubric below

*4. Predictability Gain Rubric*

**5 (Excellent Predictive Power):** The response provides a clear, generalizable principle or rule about the agent's behavior. A user would feel very confident applying this rule to predict actions in many new and different situations

**4 (Good Predictive Power):** The response provides a useful insight or pattern that could be applied to similar future situations. A user would feel reasonably confident in making predictions

**3 (Some Predictive Power):** The response hints at a general strategy but does not state it clearly, requiring the user to interpret heavily. It offers more than a simple description but is not a clear, actionable rule

**2 (Minimal Predictive Power):** The response only explains the current action in a way that is highly specific to the current state. It offers little to no insight that could be generalized to other situations (e.g., "It attacked the skeleton because it was there.")

**1 (No Predictive Power):** The response is confusing, irrelevant, or simply describes the environment without providing any reasoning. It gives the user no basis for predicting any future actions

*5. Input and Output Instruction*

You will be provided with a 'User Query' and a 'Final Response'. Your output MUST be a single integer from 1 to 5 and nothing else. Do not provide any reasoning, explanation, or additional text

*Your final output must be only one character: "1", "2", "3", "4", or "5".**

Figure A6: Evaluation prompt for Informativeness.

914
915
916
917

You are a meticulous and impartial AI assistant serving as an expert evaluator. Your task is to assess one specific criterion: **Query Relevance**.

*1. Context**

The response you are evaluating is generated by an AI "Curator" that explains the behavior of a Reinforcement Learning (RL) agent in the game "Crafter". Users ask questions about the agent's decisions, and the Curator provides an explanation.

*2. Evaluation Goal**

Your single objective is to determine how well the 'Generated Response' directly answers the 'User Query'. You will assign a score from 1 to 5 based *only* on the relevance rubric below.

*3. Evaluation Steps**

1. Read the 'User Query' to understand the user's exact intent.

2. Read the 'Generated Response'.

3. Compare the response directly against the query to judge its relevance.

4. Choose a single integer score from 1 to 5 that best represents the relevance.

*4. Query Relevance Rubric**

**5:** The response directly and completely answers the user's question without any unnecessary information.

**4:** The response accurately answers the user's question but may contain minor irrelevant details.

**3:** The response addresses only a part of the user's question or provides an incomplete answer.

**2:** The response is on the same general topic as the query but fails to answer the core question.

**1:** The response completely ignores the user's question and is unrelated.

*5. Output Instruction**

You will be provided with a 'User Query' and a 'Generated Response'. Your output MUST be a single integer from 1 to 5 and nothing else. Do not provide any reasoning, explanation, or additional text.

*Your final output must be only one character: "1", "2", "3", "4", or "5".**

Figure A7: Evaluation prompt for Query relevance.

You are a meticulous and impartial AI assistant serving as an expert evaluator. Your task is to assess one specific criterion: **Fluency**.

*1. Context**

The response you are evaluating is generated by an AI "Curator" that explains the behavior of a Reinforcement Learning (RL) agent in the game "Crafter".

*2. Evaluation Goal**

Your single objective is to evaluate the linguistic quality of the 'Generated Response'. You will assess its grammar, structure, and clarity, assigning a score from 1 to 5 based *only* on the fluency rubric below. **Crucially, the response must be in natural, human-readable language. Responses consisting of raw data, code, or unformatted lists should be heavily penalized.** The relevance of the response to any query should be ignored.

*3. Evaluation Steps**

1. Read the 'Generated Response' carefully.

2. Analyze its grammatical correctness, clarity, and overall readability.

3. Determine if the response is presented as natural language.

4. Choose a single integer score from 1 to 5 that best represents its linguistic fluency based on the rubric.

*4. Fluency Rubric**

**5:** The response is perfectly written. It is grammatically correct, well-structured, clear, and uses natural language.

**4:** The response is well-written and easy to understand, with only very minor errors that do not impact readability.

**3:** The response is generally understandable but has noticeable grammatical errors or awkward phrasing.

**2:** The response is difficult to read due to significant grammatical errors or unnatural language. **This score should also be used if the response is not primarily natural language (e.g., a raw list of keywords, unformatted data).**

**1:** The response is grammatically incorrect, nonsensical, or unreadable. **This score must be used if the response consists entirely of non-natural language content like a code block, a JSON object, or a stack trace.**

*5. Output Instruction**

You will be provided with a 'User Query' and a 'Generated Response'. You must evaluate the fluency of the response only. Your output MUST be a single integer from 1 to 5 and nothing else. Do not provide any reasoning, explanation, or additional text.

*Your final output must be only one character: "1", "2", "3", "4", or "5".**

Figure A8: Evaluation prompt for Fluency.