# OpenReview forum: "Towards Faithful Agentic XAI"
_ICLR.cc/2026/Conference — ICLR 2026 Conference Withdrawn Submission_

### Official Review · Reviewer_FC6S · 2025-10-29

**Soundness:** 3
**Presentation:** 3
**Contribution:** 2
**Rating:** 4
**Confidence:** 3

**Summary:**

This study proposes an explainable AI agent named “FAX”, which can explain actions of a RL agent (LLM) trained on “Crafter, an open-world RL environment”. Inspired by earlier studies that rely on LLMs’ capability to interact with tools, FAX, an agentic LLM, is trained to use one of 4 XAI tools to craft responses to user queries. Thus, this study is in line with earlier studies using LLMs to evaluate RL agents, but it extends this line of studies by training FAX to follow an agentic 6-step-workflow, which includes a verification step.

**Strengths:**

FAX is trained to verify its own explanation by editing a state and simulating agent’s actions in response to the edited state. With an additional verification step, FAX shows notable improvements in terms of faithfulness.

**Weaknesses:**

1. The authors test FAX in a simple artificial environment where states can be easily manipulated (via state-editing in this study), but if LLM agents are trained on more complex problems or environments, FAX verification step may not be feasible. Thus, I am not sure that FAX can be used for general RL agents.
2. Except faithfulness, the advantage of FAX to other LLM-based solutions is not prominent.

**Questions:**

1. Can the authors provide some insights why the structured XAI without verification performs better than FAX for informativeness, Query Relevance and Fluency?

---

### Official Review · Reviewer_Vxfw · 2025-10-29

**Soundness:** 3
**Presentation:** 3
**Contribution:** 1
**Rating:** 2
**Confidence:** 3

**Summary:**

The paper proposes two main things, a workflow for faithful Agentic XAI systems called FAX, and an evaluation framework called CFAFTER-XAI-Bench. The prior introduces a verification stage in the agent's workflow, and the latter primarily uses model simulation to assess faithfulness of the final explanations from the XAI agent.

Experiments showed their method had a large increase in explanation faithfulness, but no significant difference in Informativeness, Query Relevance, or Fluency.

**Strengths:**

I think it's good the paper focuses on Agentic XAI systems, it's a new and interesting area with potentially large impact. I also think that using LLMs as a judge for evaluation is timely and smart, although one should be careful to not hack the results in doing so through prompts etc. The paper also is quite accurately written, the claims in the abstract match the paper's contributions as described.

**Weaknesses:**

There are several points I feel are worth considering here:
1. The paper's usage of model simulation to evaluate faithfulness I do not believe is sufficient. It is possible to provide explanations which aid model simulation but are not actually faithful to the model. Hence, the central claim of the paper is flawed and misleading.
2. The core novelty of the paper, to add a verification stage in the Agentic workflow, is in my humble opinion not a large enough contribution for publication. It is not surprising to me that adding an additional stage for this kind of verification would marginally improve a metric like model simulation. I am unsure if this is really a research question, it reads more like an engineering solution pipeline.

Minor points
* You don't define exactly what an "agent" is anywhere in the paper as far as I can see.
* The word "confirmed" on line 101 is too strong.

**Questions:**

Can you please elaborate in detail why this is a technically novel approach? There are many faithfulness tools published, is this method simply plugging them into a prompting pipeline?

Can you please provide concrete evidence that model simulation directly proves improved faithfulness of an explanation? Demonstrate a causal link please which generalizes.

---

### Official Review · Reviewer_imFd · 2025-10-31

**Soundness:** 2
**Presentation:** 3
**Contribution:** 1
**Rating:** 4
**Confidence:** 3

**Summary:**

The paper proposes Faithful Agentic XAI (FAX), an extension of structured Agentic XAI that adds a verification step to make the explanations more faithful rather than just plausible. Instead of blindly trusting whatever XAI tool output the agent gets, FAX runs a six-stage workflow where the initial explanation draft is actually cross-checked using what they call “inherently faithful” XAI methods (like counterfactuals, state edits, feature importance). It’s kind of like giving the LLM a sanity check loop so it doesn’t hallucinate explanations.

They also introduce CRAFTER-XAI-Bench, a reinforcement learning–based benchmark designed to test explanation faithfulness in more dynamic, realistic environments, and an LLM-based simulation metric that quantifies how well an explanation predicts model behavior. Results show that this verification process boosts faithfulness scores quite a bit (around 2× over structured Agentic XAI) while still keeping explanations fluent and relevant. Overall, it’s a neat and pretty practical idea to make agentic XAI systems a bit more trustworthy, even if the step itself is conceptually simple.

**Strengths:**

The paper addresses a clear and timely issue in Agentic XAI—models that generate fluent but unfaithful explanations—and proposes a simple, effective solution. The addition of an explicit verification loop is well-motivated and implemented through a clear six-stage workflow. The presentation is strong: the writing is clear, the figures are intuitive, and the overall narrative is easy to follow despite the conceptual complexity. The introduction of CRAFTER-XAI-Bench moves the evaluation beyond toy datasets and provides a realistic setting to test faithfulness. Quantitatively, the gains are substantial (around 2× improvement over strong baselines) and are achieved without degrading other key metrics.

**Weaknesses:**

The contribution is somewhat incremental conceptually. The verification stage, while effective, functions primarily as an additional structured prompt rather than a fundamentally new mechanism. The notion of “faithful tools” is loosely defined, and the evaluation relies on a small, curated set of XAI methods, limiting claims of generality beyond the RL environment. The dependence on LLM-as-a-judge metrics raises concerns about evaluation validity, as the same model family is involved in both explanation and assessment. The absence of human or independent verification weakens the empirical grounding. Finally, the paper does not discuss computational cost or scalability, and the six-stage pipeline may introduce significant overhead. Overall, the approach works well within its scope, but its robustness and applicability to broader domains remain uncertain.

**Questions:**

How is faithfulness formally defined beyond the simulation metric, and how do you know the verification step improves it rather than just consistency?

What criteria define the “faithful tools,” and can this approach generalize beyond the specific XAI methods used?

Since evaluation uses LLM-as-a-judge, how do you avoid bias when the same model family both explains and evaluates?

What are the runtime and computational costs of the six-stage workflow? Is it practical for larger or real-time systems?

---

### Author Response · Authors · 2025-11-21

To the Area Chair and Reviewers,

We sincerely appreciate for the time and effort you dedicated to reviewing our paper. After carefully considering the valuable feedback provided by all three reviewers, we have decided to withdraw our submission to incorporate your suggestions and further strengthen our work for future submission to another venue.

Below is a summary of the feedback and our plan for improvements:

### Summary of Strengths
We are encouraged that the reviewers recognized the value of our work in several key areas:
1. Timeliness and Importance: Reviewers agreed that addressing unfaithful explanations in Agentic XAI is a critical and timely issue.
2. Effectiveness of Verification: The proposed six-stage workflow was acknowledged for delivering substantial quantitative improvements in faithfulness without sacrificing fluency.
3. Importance of the proposed benchmark: The introduction of the CRAFTER-XAI-Bench were highlighted as positive contributions.

### Addressing Concerns & Future Plan
We have taken note of the major concerns raised and have established a clear plan to address them:

1. Faithfulness Metric & Simulation: We acknowledge the concern regarding whether model simulation performance sufficiently proves faithfulness. We realize our original manuscript did not clearly articulate the causal link between simulation outcomes and the model's internal decisions. In our revision, we will rigorously rewrite this definition to clarify how the metric guarantees alignment between the explanation and the model's actual behavior.
2. Technical Novelty & Generalizability: We understand the critique regarding the incremental nature of the verification step and its limitation to the Crafter environment. We plan to conduct additional experiments in broader domains (e.g., NLP or Vision) to demonstrate that our FAX framework is a generalizable research contribution, not just an engineering solution for a specific RL task.
3. Evaluation Bias & Cost: We will also address the concerns about LLM-as-a-judge bias and add a detailed computational cost analysis.

Thank you again for your constructive feedback. Your feedback has provided a clear roadmap for evolving our research into a more robust contribution.

---

### Note · Authors · 2025-11-21

I have read and agree with the venue's withdrawal policy on behalf of myself and my co-authors.